# Measuring Micrometers of Matter and Inventing Indices: Entangling Social Perception within Discrete and Continuous Measurements of Air Quality

**Edwin Schmitt**

Department of Culture Studies and Oriental Languages, University of Oslo, 0315 Oslo, Norway;
e.a.schmitt@ikos.uio.no

**Abstract:** Environmental protection agencies around the globe are establishing different methods for measuring particulates, and then integrating those measurements into a single air quality index with other pollutants. At the same time, scientific inquiry has also shifted to a theory of measurement that incorporates discrete and continuous measurement. This article reviews the relationship between discrete measurements and indices, while also speculating on the way that the continuous measurement of air pollution could stimulate awareness and action. The paper argues that continuous measurement must include the way people of different backgrounds perceive air pollution in their lives. After reviewing the methods of measuring particulates and their inclusion into various indices, the article argues that in order to take action to mitigate the health impacts of air pollution, we must allow for the social perception of air pollution to become entangled within our scientific measurements.

**Keywords:** environmental monitoring; air quality index; atmospheric science; measurement theory; participatory science; particulate matter

---

## 1. Introduction

Generally, when the leader of a superpower references the weather during a prominent speech given at a gathering of world leaders, we assume it is nothing more than nervous pleasantry. When President Xi Jinping referenced the blue skies that emerged during the Asia-Pacific Economic Cooperation (APEC) 2014 forum hosted in Beijing, he was communicating a great deal of information. By shutting down factories two weeks before the forum began and enforcing automobile restrictions, the People's Republic of China (PRC) was able to ensure that the gray, polluted skies that had become commonplace for Beijing residents would give way to blue skies, what online commentators described as APEC Blue. Xi Jinping reflected on the weather in his speech by directly appropriating the following commentary: "I hope, and I believe through hard work, that APEC Blue can remain in Beijing" (Zheng 2014). Through this appropriation, Xi Jinping cleverly showed that he was listening to the concerns of Beijing residents, as well as the owners of corporations and state-owned enterprises. The PRC government has taken similar steps to curb air pollution during a number of mega-events, such as the 2008 Olympics and the 2016 G20 Summit (Shen and Ahlers 2019). This event can be contrasted with some of the highest levels of pollution ever measured in China in January 2013, this time dubbed Airpocalypse by several foreign media outlets (Wong 2013). Naturally, leaders of the PRC avoided this term, but state-controlled media outlets critically reviewed the event. A few weeks later, Prime Minister Li Keqiang declared a "war on pollution", and the Ministry of Environmental Protection (MEP) released its Air Pollution Prevention Scheme (Feng and Liao 2016).

These two events provide us with an opportunity to critically reflect upon how we, as a society, determine whether or not our environment is polluted, and what we plan to do about it. For instance,

Beijing residents made the determination that APEC Blue was a clean environment and Airpocalypse a polluted environment, based on their visual perception of the atmosphere, but also on tactile sensory perception, such as through breathing the air or feeling the layers of dust that accumulated in one's home. These two events also provided atmospheric scientists, armed with the latest technological instruments, an opportunity to refine the way we measure and calculate air pollution, with a particular concern for public health. As I will show in this paper, the convention of describing these two ways of determining environmental quality as distinct and separate processes is being challenged, resulting in innovative ways of measuring physical reality as well as how we represent those measurements to the public.

A productive way to think of these differences is according to discrete and continuous ways of measuring phenomenon. The purpose of this paper is, first, to provide an introduction for the way particulate matter is measured discretely and continuously by atmospheric scientists, as well as how these measurements are related to public health concerns of exposure to high levels of particulates. Second, I will review the way discrete measures are integrated into air quality indices for stimulating social awareness in the USA, PRC, and the EU. Then, I will speculate on the way continuous measurements could be integrated with two-way modes of communication found in new media. I will conclude by arguing that an entanglement of discrete and continuous measurements with social perception within a kind of participatory science is necessary for moving us beyond awareness towards changing everyday practices that could reduce air pollution.

## 2. Measuring Particulates: Discrete and Continuous

Inquiries into the scientific nature of measurement have shifted from the metaphysical to the anti-metaphysical, and in recent years, into relativistic realms (Mari 2003); a progression that has arguably contributed to the development of post-normal science (Funtowicz and Ravetz 1994). Relativistic interpretations of measurement can also be traced to early debates about how we come to make "precise measurements" in light of quantum mechanics (Wigner 1964). Max Born explained how the theory of measurement was challenged after "[Niels] Bohr . . . introduced the idea of complementarity to express the fact that the maximum knowledge of a physical entity cannot be obtained from a single observation or a single experimental arrangement, but that different experimental arrangements, mutually exclusive but complementary, are necessary" (Born 1954, p. 146). Thus, the subjectivity of the observer became recognized as an influence on the measure of a phenomenon, so that the "boundary between the action of the subject and the reaction of the object [was] blurred" (ibid, p. 147). This should not be taken to mean that accurate measurement cannot be made even at the macroscopic level (Sen 2010), but rather that measurements taken from multiple perspectives can help us to improve our accuracy. Tal (2017) describes "epistemic" accuracy as, paradoxically, a representation of a "true value" of what is being measured by looking at the convergence of multiple inaccurate measurements. Adding multiple perspectives not only increases a study's ability to collect more measurements that then converge upon a better representation of accuracy, but each perspective added to a study also ensures that the measurements made within the study are a closer representation of a socially or intersubjectively accepted process, which supports calls for engaging in post-normal science (Funtowicz and Ravetz 1994). Moreover, as each perspective becomes a distinct act of measurement, increasing the number of perspectives actually strengthens the robustness of our study through the comparison of the different measurement processes taken into account (Tal 2017).

Similarly, as I will show in this article, even while analyzing the world of tiny particles in the vast atmosphere, relativistic interpretations of measurement have not prevented scientists from still attempting to use apparatus to make more robust measurements. More importantly, I will build upon Karen Barad's concept of agential realism (2007) to argue that the integration of citizen participation within the practice of measuring air quality could have practical implications for stimulating social actions that will reduce air pollution. In this sense, the article will disagree that a measurement

should only include the "measured thing" so as not to include the environment and the observer (Mari 2003, p. 28). Instead, I will argue that today, a measurement can include both and still be capable of being communicated inter-subjectively. I consider this paper then to be a conceptual contribution to measurement science that elaborates on the social importance of the relationship between discrete and continuous measurements. How a measurand can be described as discrete or continuous is closely related to the object of its measure. Generally, discrete measures are calculated in relation to a given interval of another measure (i.e., a time interval), while a continuous measure is calculated in direct relation to another measure (i.e., instantaneity). Below, I will demonstrate that the way atmospheric scientists measure particulates in the air is an excellent example of how science itself has moved from strictly discrete measurements towards the integration of discrete and continuous measurement collected by so-called "experts" as well as "laymen".

As a social scientist, I should also note that we too have long pondered on the issue of discrete versus continuous measurement. We have often debated the need for indices that can express the way society perceives their environment in a simplified manner, not unlike the goal of the air quality indices discussed below. For instance, Cicourel (1964) noted that the modal breakdowns in a five-point scale determined by a researcher may not accurately correspond to the cultural model that our informants use in their everyday life, leading to theoretical conclusions that are not easily defensible. In this sense, he was not advocating for the dismissal of discrete measurement in social science, but rather that it be used appropriately. The way society perceives air quality has primarily been demonstrated using discrete measurements, such as scales of risk perception of air pollution (Johnson 2003). More recently, a socio-cultural interpretation has drawn from what would be better described as continuous measurement, usually obtained through historically situated ethnographic description (Bickerstaff 2004). In this article, I want to combine such studies with Science and Technology Studies (STS) research on how air pollution is measured (Mallard 1996, 2001). Moreover, by examining the shift towards continuous measurement in atmospheric science, we can see that the discipline is developing in a way that more closely mimics social perceptions of air quality, which explains the rising importance of participatory science in our understanding of the atmosphere (Bailey et al. 1999).

Atmospheric particulate matter, also called aerosols, particulates, or just PM, are tiny pieces of solid or liquid matter that are suspended in the air. Particulates come in a range of sizes, from 0.001 to 100 micrometers. For instance, in recent years, the often discussed $PM_{2.5}$ is simply a unit of measure designating fine particles that are smaller than 2.5 micrometers (Seinfeld and Pandis 2006, pp. 350–53). Because the direct measurement of an individual particle is not practical, the two most common equivalents used are aerodynamic diameter, which uses filtration and is dependent on the properties of airflow and particle density, and light scattering diameter, which uses an optical array that is dependent on the properties of a refractive index for the scattering of light (Tiwary and Colls 2010, pp. 59–60). Both methods can be used to make discrete and continuous measurements. In this article, when discussing measurements of particulate matter, I define discrete measures as measurements that are time-dependent, and continuous measures as those that are generally estimates in real-time. Other than the dimensions of particulate matter, the measurement of these particles can also be approached from two other perspectives, concentration and chemical composition, both of which have important bearing on the way toxic exposure to particulate matter is defined. A brief discussion of my experience while participating in a study on particulate matter can help place these practices into perspective.

The primary concern for that project was focused on transboundary aerosols emitted from China bringing pollution across the Pacific Ocean along the jetstream and settling in the Northwest United States and Western Canada (Lafontaine et al. 2015). In order to determine if air pollution from China was truly impacting populations in North America, the team needed to collect a series of samples from the top of Mount Bachelor, a 2764 m dormant volcano, where the sampling machines were kept. The sampling machine used in this project was a high-volume cascade impactor (Series 230, Tisch Environmental, Cleves, OH, USA) that collected 43 specimens of particulates on quartz fiber filters

between March and May in 2010 and 2011. Members of the research team would take turns going to the top of the mountain to change the filters, and each time the researcher would need to spend time calibrating the airflow of the sampling machine. In this project, the concentration was primarily determined by the weight of the filter with the particle density being a function of the rate of airflow and the amount of time that one filter spent on the sampling machine (Tiwary and Colls 2010, pp. 59, 165). This methodology is useful for discrete measures of concentration that are highly accurate over specified time intervals. Particles can also be measured using optical technologies, such as with a direct-reading aerosol monitor (DRAM), commonly found in hand-held air sampling devices, which scatters near-infrared light particles across a lens that then translates the amount of light blocked from the particles into a mass concentration reading (Tiwary and Colls 2010, p. 166). This technology is useful for determining continuous levels of particulate matter concentrations in real-time.

How discrete and continuous measurements are utilized to determine concentration is closely related to the research question being asked, and, more importantly, the way that toxicity is defined. For instance, the World Health Organization (WHO) defines unhealthy long-term exposure to $PM_{2.5}$ as $10 \mu g/m^3$ for the annual mean (WHO 2006), which is based on the reanalysis of the Harvard Six Cities Study (HEI 2000; Pope et al. 2002), which collected 24-h discrete samples similar to those collected on Mount Bachelor. A methodological treatment of the Harvard Six Cities Study was published (Samet et al. 2000a), but does not provide a detailed explanation of the sampling procedure because it is focused on how to compensate for error using mathematical functions.

The WHO definition of unhealthy short-term exposure for $PM_{2.5}$ is set at a $25 \mu g/m^3$ for a 24-h mean, which is based on the findings of research in Europe (Katsouyanni et al. 2001) and the U.S. (Samet et al. 2000b). Katsouyanni et al. (2001) collected the data from 29 European cities, which, only since the mid-1990s, were required to use a standardized methodology for measuring $PM_{10}$. In fact, the methodological standards proposed by the EU (EC 2010) are focused primarily on the mathematical modeling used to calculate concentration, and ensure that calibration is performed correctly. There could have been a number of sampling machines used, because as stated by the Commission of the European Communities, "A Member State may use any other method which it can demonstrate gives results equivalent to the above method or any other method which the Member State concerned can demonstrate displays a consistent relationship to the reference method" (Commission of the European Communities 1999, p. 59). Ultimately, the methodology for all of the studies used to support the WHO definitions of unhealthy long-term and short-term exposure could be a bit more explicit.

What we can tell from these studies, though, is that continuous monitoring was utilized and correlated with hospitalization rates for various diseases that are known to either be caused by or exacerbated by exposure to particulate matter (see below). In the end, the WHO standard was determined based on the methodologies that utilized a filter-based gravimetric method rather than an optical technology, and incorporated one five-year study based on 24-h samples of discrete measurements (Katsouyanni et al. 2001), and one seven-year study based on 1-h samples of discrete measurements (Samet et al. 2000b). However, as we will see, there is a general shift towards incorporating continuous measurements using optical technologies with discrete measurements that use filter-based gravimetric methods for determining concentration, which is a result of air pollution research becoming more embedded within participatory science frameworks, and could come to influence the way standards are determined at institutions like the WHO.

The filters that are collected in aerodynamic diameter methods for calculating particulate size and concentration can also be used to determine the chemical composition of the particulates. Once such filters are properly prepared using dichloromethane as a solvent to remove the particulates from the quartz fiber filters in a process called pressurized liquid extraction (Primbs et al. 2008), the samples are then analyzed via gas chromatography–mass-spectrometry (often shortened as GC/MS), in order to determine their molecular composition. GC/MS is actually two technologies utilized in tandem. The first, gas chromatography is similar to the process of distillation, separating out a number of different molecules that are contained within a properly prepared glass tube, called a column. The second,

mass spectrometry, utilizes the physical quantity of the mass-to-charge ratio, which states that each unique molecule has a correspondingly unique mass and electric charge that helps determine the particle's motion through time and space. After the gas chromatographer separates the different molecules in a column, the mass spectrometer then bombards them with ions and accelerates them towards an electron multiplier, which picks up the ions that are deflected away from the material being tested. This is then recorded as a spectra, and can be interpreted according to the spectra of other known materials (Sparkman et al. 2011). Here is where the process of calibration is closely related to what Mallard describes as "authoritative references" or "certificates of approval, labels, or letters of recommendation" (2001, p. 234). For instance, Lafontaine et al. (2015) purchased standards of the chemicals that they were studying in order to compare their spectra against the samples they had collected at Mount Bachelor. In this case, the apparatus was an Aligent 6890 gas chromatograph (Santa Clara, CA, USA) coupled with an Agilent 5973N mass selective detector (Primbs et al. 2008). The Aerodyne Aerosol Mass Spectrometer is another technology that utilizes particle beams to determine both the size and chemical composition of particles in real-time (Peck et al. 2016) although this and similar technologies are still being field-tested (Zhang et al. 2017).

While the composition of these particulates can include materials such as fungal spores or other organic compounds, it is the impact of toxic components like polycyclic aromatic hydrocarbons (PAHs) that most concerns atmospheric scientists and toxicologists, which is why Lafontaine et al. (2015) chose it as the focus of their study. PAHs are a form of particulate matter that include various combinations of carbon and hydrogen-based organic compounds produced through incomplete combustion, and are composed of at least seven benzene rings (Seinfeld and Pandis 2006, pp. 670–675). PAHs with an increasing number of these benzene rings have been demonstrated to have an increasing level of carcinogenicity (Boström et al. 2002). Thus, in China, some researchers have focused on how certain policies to limit the production of particulate matter, such as during the 2008 Olympics, have actually served to reduce hospital visits or even lower instances of cancer rates in the short-term (Jia et al. 2011). Moreover, after the so-called Airpocalypse in 2013, researchers were able to show that there was an increased risk for adults in China to be diagnosed with cancer when exposed to PAHs (Lu et al. 2015).

Beyond the danger that the chemical composition of PAHs pose for human health, particulates can increase instances of heart disease, as it has been determined that such materials can enter the bloodstream through the lungs (Brook et al. 2010), or asthma, through long-term exposure (Koenig 1999). Pope et al. (1991) discovered that exposure to higher levels of $PM_{10}$ resulted in decreased lung function, which led to numerous epidemiological studies regarding the relationship between asthma and particulates in Europe (Sunyer et al. 1997), the United States (Sheppard et al. 1999), and China (Lu et al. 2013). As Jones (1998) noted in the mid-1990s, there was still a great deal of uncertainty regarding the relationship between asthma and outdoor air pollution. There was also a large interest in the relationship between asthma and exposure to indoor air pollution. The literature on indoor air pollution and health is now quite extensive (Brown et al. 2010), and while there are certainly important overlaps, here, I focus primarily on outdoor air pollution, because of its importance to the creation of indices.

Asthma, however, is a very complicated disease, because there is evidence that genetics play a part in the emergence of asthmatic symptoms as a result of exposure to various environmental factors at a young age (Childhood Asthma Management Program et al. 2011). Some of these epidemiological studies showed that the Latino population exhibits lower instances of asthmatic symptoms (Homa et al. 2000; Moorman et al. 2011). As a result, government programs in the United States that help asthmatics cope with their disease are either reduced or cut in Latino communities, despite the fact that many of those communities are prone to a high exposure of air pollutants (Schwartz and Pepper 2009; Grineski 2007; Grineski et al. 2013). It is precisely for this reason that information regarding the relationship between exposure to air pollution and human health needs to be made more relevant to the needs of individual communities. Such information becomes even more difficult to keep in the foreground when data on air pollution becomes integrated into the indices I will describe below.

## 3. Inventing Indices of Air Pollution

Incorporating the measure of multiple air pollutants into a single index is a long-standing environmental management practice, and even at the birth of such indices, it was recognized that particulate matter was the "most serious air pollution problem" (Babcock 1970, p. 658). Shooter and Brimblecombe (2009) have argued that an Air Quality Index (AQI) is necessary to transmit information about the health effects of air pollution to the public effectively, and to support the development of proper policies to reduce air pollution. Similarly, I wish to make the rest of this article about inventing indices for "drawing the public's attention to air quality issues and raising awareness" (Van den Elshout et al. 2014, p. 462). First, I would like to review the way the United States, China, and the EU currently formulate AQI, so that we can evaluate the practicality of introducing new tools for stimulating social awareness of the problem of air pollution.

Since the formation of the Clean Air Act in 1977, the United States Environmental Protection Agency (U.S. EPA) established the national ambient air quality standards (NAAQS). These standards were originally used to calculate the Pollution Standard Index (PSI), using a segmented linear function that was defined by a series of breakpoints, based on the scientific evidence that air pollution was harmful to human health (Ott and Hunt 1976). Through a series of regular reviews (generally every fifth year) supported by input from the public and the Clean Air Scientific Advisory Committee (CASAC), updates to the NAAQS standards are made according to the advancement of scientific knowledge (Jordan et al. 1983; Padgett and Richmond 1983; Jasanoff 1990, chp. 6). The PSI was revised in 1999 by including subindices for $PM_{2.5}$ and $O_3$, and was renamed the Air Quality Index. Cheng et al. (2007) demonstrated that by including $PM_{2.5}$, AQI provided a more accurate index than PSI. This regular updating of standards makes the formation of the AQI quite complicated, because as Elshout, Léger, and Heich rightly argue, these "are communication tools so people have to get used to them and frequent changes are not recommended" (Van den Elshout et al. 2014, p. 461). For this reason, it is not a simple matter for the EPA to update any adjustments to the index, because the linear interpolation formula (Equation (1)) used to calculate AQI is dependent on Table 1, which was established by the EPA according to NAAQS recommendations.

$$I_p = \frac{I_{Hi} - I_{Lo}}{BP_{Hi} - BP_{Lo}}\left(C_p - BP_{Lo}\right) + I_{Lo} \tag{1}$$

$I_p$ = the index value for pollutant p

$C_p$ = the truncated concentration of pollutant p

$I_{Hi}$ = the AQI value corresponding to $BP_{Hi}$

$I_{Lo}$ = the AQI value corresponding to $BP_{Lo}$

$BP_{Hi}$ = the breakpoint that is greater than or equal to $C_p$

$BP_{Lo}$ = the breakpoint that is less than or equal to $C_p$

**Table 1.** Air Quality Index (AQI) levels and category for calculating $PM_{2.5}$ subindex adapted from the Environmental Protection Agency (EPA 1999). Each color conforms to a given category.

| $PM_{2.5}$ µg/m$^3$ | AQI | Category |
|---|---|---|
| 0.0–15.4 | 0–50 | Good |
| 15.5–40.4 | 51–100 | Moderate |
| 40.5–65.4 | 101–150 | Unhealthy for sensitive groups |
| 65.5–150.4 | 151–200 | Unhealthy |
| 150.5–250.4 | 201–300 | Very Unhealthy |
| 250.5–350.4 | 301–400 | Hazardous |

The final AQI is determined according to which subindex calculates the highest AQI. The EPA includes subindices of $PM_{2.5}$, $PM_{10}$, $O_3$, CO, $SO_2$, and $NO_2$ (but the latter only once it reaches 0.65 ppm).

As I have focused on particulate matter here, I will use the $PM_{2.5}$ reading of 26.9 μg/m$^3$ here in Hong Kong on 23 August 2016 (where I am currently writing) as an example. Following Table 1, we first need to find the breakpoints where this reading would fit (15.5–40.4), and then the max and min levels of AQI for this subindex (51–100), allowing us to calculate an AQI of 73, or a "moderate" level of air pollution caused by $PM_{2.5}$. In fact, the ozone levels today are in the "unhealthy for sensitive groups" range (91.8 μg/m$^3$), so the actual AQI was not defined by $PM_{2.5}$. We can already begin to see how this can become quite complicated, as collections of discrete measures of different pollutants are aggregated into a single index. This method also obfuscates the fact that exposure to each type of pollutant will result in health impacts.

Because the EPA established this formula and set of standards early on, many countries around the world have simply adapted the use of AQI, but often with various caveats. The PRC, for instance, has changed the way the breakpoints between the "good" to "unhealthy" categories are structured (Table 2). They also have added specific standards for $NO_2$ that are not used in the US system. This makes sense because $NO_x$ has been a fairly persistent problem in China due to the inefficient burning of coal for power generation, which some scholars expect will become a larger problem in the future (Zhao et al. 2008).

**Table 2.** AQI levels and category for calculating $PM_{2.5}$ subindex adapted from the Ministry of Environmental Protection (MEP 2012). Each color conforms to a given category.

| $PM_{2.5}$ μg/m$^3$ | AQI | Category |
|---|---|---|
| 0.0–35 | 0–50 | Good |
| 36–75 | 51–100 | Moderate |
| 76–115 | 101–150 | Unhealthy for sensitive groups |
| 116–150 | 151–200 | Unhealthy |
| 151–250 | 201–300 | Very Unhealthy |
| 251–350 | 301–400 | Hazardous |

In the PRC, the current levels of $PM_{2.5}$ in Hong Kong would calculate out an AQI of 38, which is solidly in the good category. Lou (2016) has explained that Hong Kong residents feel a sense of foreboding when it comes to air pollution, because it is viewed as floating into the city from Mainland China to the north, which mirrors their concern for Mainland Chinese political influence in Hong Kong. This could be one reason Hong Kong has established a completely different way for calculating an index, which they describe as the Air Quality Health Index (AQHI). The AQHI is based on an index created in Canada (Stieb et al. 2005), which combines local health records with specific air pollution events to establish what the Hong Kong system describes as an "added health risk" (%AR) factor. The %AR was determined using a database of respiratory and cardiovascular disease hospital visits over the last five years, and correlated the results of these visits to exposure to air pollution. The WHO standards were used as a baseline, and then five-bands and a ten-point index for the AQHI (Table 3) were established according to a given %AR.

**Table 3.** "Added health risk" (%AR) factor levels for calculating the Air Quality Health Index (AQHI) and categories adapted from Wong et al. (2012). Each color conforms to a given category.

| %AR | AQHI | Category |
|---|---|---|
| 0.1.88 | 1 | Low |
| >1.88–3.76 | 2 | Low |
| >3.76–5.64 | 3 | Low |
| >5.64–7.52 | 4 | Moderate |
| >7.52–9.41 | 5 | Moderate |
| >9.41–11.29 | 6 | Moderate |
| >11.29–12.91 | 7 | High |
| >12.91–15.07 | 8 | Very High |
| >15.07–17.22 | 9 | Very High |
| >17.22–19.37 | 10 | Very High |
| >19.37 | 10+ | Serious |

This model is fundamentally different from that used in the United States or PRC, because it is an aggregate index of %AR for each of the pollutants that are measured in Hong Kong. This includes $O_3$, $SO_2$ $NO_2$, and either $PM_{2.5}$ or $PM_{10}$, whichever is higher. As detailed by Wong et al. (2013); Wong et al. (2012), the function used for calculating AQHI in Hong Kong is as follows:

$$AR_x = \sum ((\beta x \times Cx) - 1) \times 100\% \qquad (2)$$

$\beta x =$ the added health risk factors of the respective pollutants
$C_x = 3 -$ hour moving average concentration in $\mu g/m^3$

Another fundamental difference is that a single calculation is not made for the geographic region of Hong Kong. Instead, each monitoring station (including three roadside stations) calculates the AQHI simultaneously, and then updates the information to a website (http://www.aqhi.gov.hk/en.html) on an hourly basis. Thus, the Hong Kong AQHI introduces the elements of health impacts, aggregation, and spatial complexity into the formation of an air pollution index.

More recently, scientists have introduced the idea of comparability into the formation of indices in Europe. The sampling and measurement of particulate matter has yet to be tightly regulated or standardized across the EU, and how these measurements are calculated into an index is equally diverse. For instance, the United Kingdom established an index that was closely connected to health effects, much like in Canada and Hong Kong. In contrast, Poland uses a function quite similar to the U.S. EPA model, except that their breakpoints are linear, and they include the pollutant benzene in their calculation. While the European Environmental Agency (EEA) does not promote a single unified index across the EU, 100 cities now voluntarily provide their data to support the CITEAIR project's development of the Comparative Air Quality Index (CAQI) (Van den Elshout and Léger 2006). The CAQI is calculated using a grid of breakpoints, using linear interpolation that follows a calculation similar to that used by the French-based L'Indice de Qualité d'Air or the ATMO index (Henry 2004). However, at no point is an actual function for calculating the index provided in any of the literature that I have reviewed (See Van den Elshout and Léger 2006; Van den Elshout et al. 2008; Van den Elshout et al. 2014).

The primary purpose of the CAQI is to provide an online framework that will stimulate awareness of the air pollution problem by allowing users to compare and contrast air pollution levels, regardless of how an AQI level is calculated in a local context. However, even its creators admit that there is no way to control for the quality of the data collected, stating that they "will take for granted whatever a city supplies as input" to their model (Van den Elshout and Léger 2006, p. 8). While the creators of CAQI may indeed have introduced the element of comparability in order to stimulate awareness, it can also provide an additional metric for determining whether or not local environmental governance is in compliance with the relevant regulations. Seeing that most of the cities that are volunteering to work with CAQI are in France, Germany, and England, where air pollution is already discussed more transparently and therefore is less politically contentious, this may introduce a political aspect of comparability that makes other cities within the EU wary about participating. Some cities in China are now also calling for utilizing the comparability of AQI as a metric for evaluating cadres at multiple levels of the government, which could stimulate a stronger enforcement of air quality standards (Feng and Liao 2016).

If we are more practical (and critical!) in our analysis of these indices, then there are several important trends that emerge. First, some indices do attempt to integrate the concern for how exposure to air pollution impacts human health. However, the functions used to calculate such indices are typically complex, introducing multiple levels of error, an issue that is already a key debate in the way the health impacts of air pollution are determined (Samet et al. 2000a; Lipfert and Wyzga 1997). Moreover, such indices must be site specific, because they are integrated with local epidemiological data (Stieb et al. 2005). This means that comparability would be limited to a contentious area that is deemed demographically homogenous, raising a number of issues regarding how environmental

health and society are understood, particularly in the genetically diverse population of large, globalized cities, where high levels of air pollution are most likely to be found.

The stimulation of social awareness is also not so straightforward. While keeping such indices simple has been a stipulation from the beginning, the functions used to calculate these indices are technical, to the point where the creators of the CAQI even state the following: "The website should assist in raising awareness; technical information relevant to subject matter specialists is generally not provided" (Van den Elshout et al. 2008, p. 721). It may be true that some of the public may well be incapable of understanding all of the technical details of how indices are calculated, but to not even provide the information seems to be working against the idea of stimulating public engagement, which could ensure that awareness of air pollution will translate into environmental practices that can help solve the problem. In an age of post-normal science (Funtowicz and Ravetz 1994), this does not have to be the case, as we should now be looking for ways to work with an "extended peer community" in the evaluation of scientific knowledge, as it impacts policy.

## 4. Entangling the Scientific Measurement and Public Perception of Air Quality

Early on, Bailey et al. (1999) showed how engaging with the public was crucial for determining the types of information that should be provided to citizens about air pollution. Because our environment and society are in constant flux, we are in need of establishing a methodology that would allow for more regular feedback from the public, which could have influence upon the way air quality is measured and then presented to the public in various forms of media. This can be done in two ways, namely: (1) encouraging greater use of small handheld air pollution monitoring devices by multiple stakeholders, and (2) utilizing social networking applications to collect public perceptions of air pollution and stimulate social action.

The use of small handheld devices has very recently become an affordable option for increasing both the temporal and spatial resolution of air pollution data (Jovašević-Stojanović et al. 2015). For instance, the light-weight Dylos 1700 (Dylos Cooperation, Riverside, CA, USA) was field-tested with 17 volunteers wearing the monitors in backpacks, and they were validated for accuracy by comparison with the stationary Tapered Element Oscillating Microbalance Filter Dynamics Measurement System (TEOM-FDMS, Thermo Fischer Inc., USA) (Steinle et al. 2015). Weighing only 1.2 pounds and costing $425, they are a practical solution for developing large-scale Citizen Science studies of particulate matter exposure. Studies of the TSI Sidepak (TSI, Inc., Shoreview, MN, USA) and MicroPEM V3.2 (RTI International, Research Triangle Park, NC, USA) have also demonstrated that they more closely approximated individual exposures than fixed monitoring systems (Sloan et al. 2016). However, it should be noted that both devices are dramatically more expensive than the Dylos 1700, and require a project manager to first request a quote from the respective sales departments at TSI, Inc. and RTI International.

Holstius (2014) has argued that as an increasing number of sensors come online among the public, it will support more timely policy decisions made by local governments regarding issues such as mitigation policies and the pricing of traffic congestion in order to reduce air pollution levels. The overall reduction in the cost of particulate matter sensors has stimulated the installation of community-based continuous monitoring systems (Jiao et al. 2015), which are necessary for a community that uses handheld devices to calibrate their equipment on a regular basis. For instance, Jiao et al. (2015) study included a pDR-1500 (Thermo Scientific, Waltham, MA, USA) within the package of instruments that they established within a community-based monitoring system. Even when such community-based monitoring systems are available, though, Holstius et al. note that "the key parameters for calibration by co-location (closeness and duration) are bounded by serious practical and logistical constraints, including scarcities of time and trusted personnel" (Holstius et al. 2014, p. 1127). Concerns over time allocation and trust are inherently social issues, and need to be negotiated within a community.

Beyond the social limitations of these sensor networks, there are also political consequences that need to be addressed. Such networks may provide the impression that citizen participation in scientific study is an example of a democratic way to influence environmental policy, and forms the proper environmental subjectivity of a citizen. However, there is no guarantee that simply collecting data would result in actions that would have any significant effect on policy or the environment, because "this would require changing the urban 'system' in which they have become effective operators" (Gabrys 2014, p. 43). Moreover, the collection of air pollution data through participatory science will only be effective if all stakeholders, including say corporate executives, agree on what it is that the instruments are actually measuring (Ottinger 2010). While Lave (2012) is pessimistic about what participatory science can do within neoliberal regimes, the Environmental Justice movement has successfully supported participatory science to gain influence over environmental policy debates (Ottinger and Cohen 2012), particularly for groups who have been socially marginalized by their race, class, and gender (Powell et al. 2011; Burke and Heynen 2014). As Janice Harper explained, heterogeneous social contexts have created differences in the way we perceive the health impacts of air pollution (Harper 2004). The measurements of air pollution from community-based continuous monitoring systems will not necessarily be able to account for these differences in perception, and in turn, will not ensure that mitigating action will be taken within local communities.

Social networking applications combined with volunteered geographic information (VGI), however, do have the potential to help communities at different spatial scales make this jump towards mitigating action. As an example, the PetaJakarta.org system utilizes Twitter posts about flooding in Jakarta to provide a real-time model of how flooding is affecting the city during the monsoon season, which is then used by the city government to help local citizens mitigate the problems (Holderness and Turpin 2016). In the context of air pollution, Okugami et al. (2014) have shown how Twitter conversations can help us understand the way particulate matter impacts suffers of asthma in real-time. However, this study is an example of one-way communication analysis. In order to stimulate social action, Holderness and Turpin (2016) sent programmatic invitations to those talking about floods on Twitter to provide additional information and link up their VGI with the PetaJakarta.org database. This allowed them to transform on-the-ground data and perspectives into visualizations that Jakarta decision-makers could use to mitigate the negative impacts of flooding. These kinds of participatory science practices are at less of a risk of turning into a "lab experiment" (Bogner 2012), and are akin to the "speculative form of participation" seen in Jennifer Gabrys's "air walks" (Gabrys 2017).

Using social media in this way could also be reinforcing to community-based continuous monitoring systems. First, a filtering algorithm similar to that developed by Mei et al. (2014) could be used to analyze discussions of air pollution and relevant key words such as APEC Blue. Then, similar to the @ErtBot studied by Wilkie et al. (2015), programmatic messages could be sent to those posting about air pollution, encouraging them to share their health concerns and upload pictures of what they consider pollution. Participants linked to community-based continuous monitoring systems could be encouraged to share their data on social media and to include a VGI stamp. Then, people who post information or concerns about air quality to social media from within a given distance could be directed to the accounts of participants collecting the continuous data, and encouraged to add them as "friends". Additionally, these automated two–way communication bots could be extremely useful for providing residents with links to more concrete information about how to mitigate the problem of air pollution through the use of face masks and air filters, as well as ways to reduce their personal contribution to the problem. Finally, such messaging could provide communities with up-to-date information that moves between citizens and decision-makers, and keeps up with the speed of the pollution, which is actually impacting the lives of citizens (Ottinger 2013). Additionally, all of this social media activity could be stored in a database and used to create a symbolic index, as discussed more in the conclusion. Once certain textual and visual patterns in the data are established, the social media database could be programmed to utilize these patterns for generating a regularly updated

symbolic index, which could include a combination of real-time particulate measurements, a word cloud, a rotating album of posted pictures of pollution, and other visualization techniques.

Even though these methodologies are still very much in the experimental phase, it is important to begin contemplating steps for implementation and examining potential pitfalls. Here, I suggest the following four initial steps:

1. Relevant environmental protection agencies should appropriate funds for supporting the establishment of community-based continuous monitoring system pilots. This should at least include funds for purchasing a monitoring system, such as the pDR-1500 or the TEOM-FDMS, for inspection and maintenance of the system, as well as for training community members who wish to participate with small-handheld monitors how to properly calibrate their devices. Funds should also be allocated to support the purchase of small hand-held devices for poor households who are interested in participating, but are unable to afford the investment. Subsequently, funds should be allocated for developing the social media aspects of the project.

2. Applied social scientists can search out communities who are actively concerned about the air quality in their region to serve as pilot projects; a concerted effort should be made to collaborate with communities who are vulnerable to particulate matter exposure. It will be necessary to determine through ethnographic inquiry the precise needs of multiple stakeholders, and then begin forming a collaborative organizational structure for establishing a community-based continuous monitoring system. This organizational structure should be flexible to local social and cultural norms, but should endeavor to recruit stakeholders who can represent the diverse ethnic, class, and gender identities in the community.

3. The community can then apply for funding to support the community-based continuous monitoring system from the relevant environmental protection agency, and begin calibration training. The community can begin setting up their social media presence and sharing their findings. The environmental protection agency can allocate a social media technician for developing a database that will collect social media posts, both textual and visual, as well as any VGI data. An applied social scientist can begin designing programmatic messages to help recruit more community members into the organization and to direct people towards information that will stimulate greater social awareness and mitigating actions.

4. In collaboration with community members, an applied social scientist and the social media technician can begin analyzing the database for information that can populate the symbolic index.

Beyond these four steps, we need to be aware of the complications that could emerge from the different roles discussed. For instance, officials in the environmental protection agency need to make sure that their requirements for funding initially are flexible, at least until several pilots have been completed. Second, social scientists can play an important role as a bridge for communicating the needs and interests of community stakeholders to officials, and vice versa. However, social scientists also need to be critically reflective about how they might be imposing their own concerns upon the development of the community-based continuous monitoring system, which may not be relevant to actual residents. Similarly, social media technicians need to work closely with social scientists and community members to ensure that their coding practices are capable of properly representing the expressions of residents, and thereby creating an accurate symbolic index of particulate matter. All coding practices should be made using open access software, and community members should be encouraged to learn how to contribute to the coding process.

While not necessarily capable of precisely mimicking what Barad (2007) calls the intra-action between the materiality of particulate matter, the environment, and humanity, the use of speculative methods (Wilkie et al. 2015) on social media has the potential to approach agentive realism in a pragmatic way. Transmitting continuous measures of air quality created by fellow citizens, and receiving responses that point us towards strategies for the mitigation and reduction of the problem, aims to add a greater number of perspectives into our analysis of air pollution and into our search for

practical solutions. Some steps of this methodology are already emerging organically. For instance, during a period of particularly bad pollution in the city of Chengdu, China, citizens shared pictures on their handheld monitoring device while standing nearby a petroleum refinery. Unfortunately, pictures of this nature also engaged in a kind of truthiness to stimulate protests against the refinery, resulting in the Chinese state shutting down the discussion of air pollution, and delegitimizing participatory practices as anti-science (Schmitt and Li under review). In other words, we need to be cautious about assuming that speculative methods are an end-all-solution. Making them effective tools for research and public participation still requires carefully crafted forms of analysis and practice in order to integrate continuous measurements of air pollution with the addition of citizens' perceptions and voices.

## 5. Conclusions

There are still many ethical issues that remain unresolved when it comes to using VGI to solve public health concerns (Goranson et al. 2013), and the use of big data for environmental monitoring collected by everyday citizens is still in the midst of being developed (See Gabrys 2016). However, this trend is a strong reflection of the changes in measurement theory and of our understanding of the environment. Discrete sampling of the air in a single location is still important for calibrating our instruments that take continuous measurements, and the latter are expanding rapidly across multiple locations. Moreover, there are even new advancements in the calculations of air quality indices that are drawing from fuzzy logic, so that the final AQI is not based on the highest subindex, but rather on a collection of inferences between the different subindices (Sowlat et al. 2011; Carbajal-Hernández et al. 2012). While this is a better way to model the health impacts from multiple pollutants, it still does not incorporate the perspectives of those affected by air pollution. The proposed method of participatory science described above is not meant to suggest that we should ignore the measurement of particulate matter and other pollutants, but is rather meant to suggest ways that such measurements could be made to enter into a more useful dialogue with social discourse, about how different groups are perceiving the problem of air pollution.

I want to stress that what I am proposing here is quite different than using social media discourse as a proxy for AQI (See Wang et al. 2017). On the contrary, speculative methods can be useful for developing an index, in the symbolic sense, that "some entity is perceived to be signaled in the context of communication incorporating the sign vehicle" (Silverstein 1976). Instead of a complex mathematical scale, the sharing of information on social media is itself an unconventional index that still signals concern about air quality, and even makes it tailored to a specific socio-geographic context. In this way, it is more truly a symbolic index, because the geographic information embedded within our social media activity recursively encourages practices of community interaction for measuring micrometers of matter, which matters to those living within said communities.

Finally, such methods of participatory science are examples of society moving away from an understanding of science as being purely objective, and finite knowledge based on the precise measurement of phenomenon. Rathindra Sen's very recent attempt to use mathematics to bridge what is described as the Heisenberg Cut, or the gap between quantum events and our conscious observation, came to the conclusion that perfectly precise measurements are impractical, which does not necessarily mean we simply give up on attempting to innovate the way we measure (2010). Today, we have to recognize that what we know about particulate matter and the apparatus that we use to measure it, is filled with indeterminacy. Long ago, Schrödinger presented this idea of measurement in terms of the "entanglement of prediction", meaning that "[i]f two separated bodies, each by itself known maximally, enter a situation in which they influence each other, and separate again, then there occurs regularly that which I have just called entanglement of our knowledge of the two bodies." (Schrödinger 1980, p. 332). It is interesting that Tim Ingold, drawing from a very different line of thought, has proposed a nearly identical definition of entanglement. According to Ingold (2015), knowledge has a blob like characteristic, but these blobs are entangled through lines (what Schrödinger would have called

"traces") that are extended from one form of knowledge to another. The measure of air pollution from one community to another may be taking on this blob like quality, but as described here, it is through social media that the lines of this knowledge are becoming entangled together across the globe. If we follow such a perspective towards its end, we would still utilize discrete measures of air pollution for calibration, but continuous measurements generated by localized users and distributed within a community will be used to demonstrate impact to public health in real-time. The sharing and re-posting of those measurements can also stimulate automated responses that keep the conversation going, while also informing citizens of solutions for mitigating health impacts and reducing our contribution to air pollution. Following Karen Barad's (2007) conceptualization of agential realism, such a system would explicitly want community members to be part of the process of conducting experiments, which could help us gain a deeper understanding of the problem from multiple perspectives.

When science is explicitly construed as being socially constructed and integrated into the process of knowledge creation, then new questions need to be asked. For instance, who gets to determine which pollutants are included within the calculation of a fuzzy AQI? Who determines which citizens are allowed to participate in recording handheld air pollution sensor data? Why those citizens? Politically-based questions of this nature are important for ensuring that environmental justice is achieved, which is not something that is readily measurable in discrete terms. The measurement of air pollution is in itself good to think with, if we consider it through the merger of both Ingold's (2015); Schrödinger's (1980) conception of entanglement. By doing so, we ensure that the measurement of the environment moves beyond an entanglement between scientists and their object of measure, and towards an entanglement with society-at-large.

**Funding:** This research received no external funding.

**Acknowledgments:** I would like thank the three anonymous reviewers, all of whom were experts in atmospheric chemistry, for their exceptionally productive comments. In particular, I was very pleased one reviewer requested I add a section detailing an action plan for implementing my ideas. Additionally, a discussion of an earlier version of this paper with Kristin Aunan was instrumental in making sure the structure of my argument was better supported by the actual practices of field researchers. Jia Yuling also helped me cut back the more impractical ideas I had for an earlier version of this paper. In the end, I am responsible for all remaining mistakes. This article would not have been possible without the financial and intellectual support of the Airborne Project and the Department of Cultural Studies and Oriental Languages at the University of Oslo.

**Conflicts of Interest:** The author declares no conflict of interest.

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
