# Peer review of "Measuring Micrometers of Matter and Inventing Indices: Entangling Social Perception within Discrete and Continuous Measurements of Air Quality"

_socsci, doi:10.3390/socsci8020048_

Round 1
Reviewer 1 Report
Being an atmospheric scientist, in particular a modeller of atmospheric composition, I read with interest the submitted paper, because it intercepts a current society need regarding a more detailed and "personal" information on air quality. The manuscript is generally quite clear and refer to adequate published literature. I have a few comments/suggestions that might be integrated in the final version of the manuscript, for which I suggest publication:
- main comment: I am confused by the definition of "discrete" and "continuous" "measurement" adopted (actually implied, since there is no definition in the manuscript) by the author. In the middle of the reading I though it was referred to the "dicrete" reporting scale of the AQI as compared to the "continuous" scale of the real measurement. However, at the end I think the author was implying that "discrete" was the official network of air quality agencies with few points, as opposed to the "continuous" network of personal monitors and social media reactions. In both cases, I think the terms are not completely appropriate: in the case of AQI I would change the word "measurement" with "scale" or "reporting scale", while in the case of the density of the network I would use "sparse" vs. "diffuse". Please clarify this point and revise the text accordingly.
- l. 89: Please expand the acronym "STS" (state of the science?)
- l. 100-101: the observation of the aerosol size is not so straightforward as implied here. I would remove this statement, it also depends on the "...the use of apparatuses, such as sampling machines, airflow dynamic modeling,..."
- l. 133: probably PM2.5 should be PM10 here
- l. 233: please change O^3 with O_3 here and throughout the manuscript. The 3 must be in the pedix.
- eq. 2: Cx or Cp? Not clear, please check.
Author Response
I want to thank all the reviewers for their very constructive critique of my article. It was very fulfilling to hear positive feedback from two atmospheric chemists and an expert on AQI who recognize that social science can also play a productive and collaborative role in the measurement of particulate matter. It is my hope that this conceptual article can help stimulate new ways of measuring particulate matter that simultaneously increases awareness and action among those exposed to this pollution.
I very much appreciate reviewer 1 for pointing out my lack of clarity in the way I use the concepts of discrete and continuous measurements. I am using these terms in the way they are understood in measurement science, but I also want the reader to understand what that means practically in terms of measuring particulate matter. Thus, in the new manuscript I have changed some language at line 45-48, and at line 80-84 I have given an abstract definition of these terms. Then again at line 114-116, I have provided a more specific definition as they are used to understand different ways of measuring particulate matter. I also have made changes to the abstract in order to clarify these issues.
Line 89 I have extended the acronym
Line 100-101 I have removed the sentence.
Line 133 This sentence is correct. The 24 hour mean for PM 2.5 is 25 μg/m3 24-hour. The details can be found here: https://www.who.int/news-room/fact-sheets/detail/ambient-(outdoor)-air-quality-and-health
Line 233 (and the other two locations): I have shifted this to a subscript.
Eq. 2: I have changed this to Cx
Reviewer 2 Report
It seems like a review of AQI at the beginning with some new proposal regarding AQI. But author/s discuss more on PM measurement. Review was not conducted properly regarding AQI. Followings are my comments: Line 21: Include full form of APEC Line 23: Include full form of PRC Line 25: Can you include references regarding these online comments? Line 89: Include full form of STS Line 111: Add space between ‘fiber’ and ‘filters’ Lines 35-36: Shouldn’t it say ‘the government’ rather than ‘we’? Preceding paragraph talks more about the government than the people. Line 42: Is it ‘I’ or ‘we’? Line 74: Add ‘pollution’ or ‘quality’ after ‘air’. Lines 100-103: ‘While the measurement of dimension is straightforward, concentration and chemical composition are inherently dependent upon the use of apparatuses, such as sampling machines, airflow dynamic modeling, and mass spectrometry.’ Please rewrite/correct this sentence. There are standard protocols and quality controls that should be followed as with any other measurement. Remove ‘airflow dynamic modeling’ as you’re talking about the measurement. Line 116: Remove ‘with the particle density being a function of the rate of airflow’. There is no information about density on the cited page (165) of this book. Line 117: Remove ‘of course’ Lines 129-132: Remove this sentence. Lines 147: 150: This sentence needs a reference. Line 148: replace ‘filtration’ with filter-based gravimetric method. Lines 150-153. They usually show the comparison with gravimetric method. Lines 154-155: remove ‘of course’ Rewrite this sentence to say chemical composition of PM can be analyzed from these filters. Lines 154-176: There are many methods and instruments to analyze chemical composition of PM. GC/MS is one of these methods. Detailed method is provided on GC/MS but it doesn’t clearly mention what it measures. Also, other methods and other instruments should be mentioned. Lines 177-178. These spores are related with several illnesses so, implying them as harmless won’t be a correct statement. Line 178: replace ‘toxic particles’ with ‘toxic components’. Line 181: Change ‘combution’ to ‘incomplete combustion’ Lines 190-192: Not all particulates can enter the bloodstream. Need reference for the statement ‘all particulates can increase instances of heart disease’. Line 193: remove ‘to the respiratory system’. Line 204: remove ‘et al.’. Line 204: Need citations for the epidemiological studies. Line 206: Need reference for the statement regarding the reduced programs for asthmatic among Latino communities. How about among other communities (African American)? Lines 209-211. This statement following the preceding sentences do not follow well. Lines 233, 245, 279. 3 should be subscript in O3. Line 250: What is ‘today’ date? Section 3. Inventing Indices of Air Pollution Why didn’t author do review regarding the pros and cons of AQI? If AQI is the focus of the manuscript, shouldn’t author discuss more on AQI than to go detail about how PM or PM chemical composition is measured? Page 7, Equation 2 description: Hyphen should be smaller after ‘3’. Line 307: ‘France, Germany and England, where air pollution is rarely an issue’. Incorrect statement. Please review the literature regarding the health effects of air pollution in these countries. Lines 313-314: First, some indices do attempt to integrate the concern for how exposure to air pollution impacts human health. Isn’t the main purpose of AQI is to warn about the health impacts of air pollution? Doesn’t that imply that the health impacts are already integrated? Lines 317-318: Moreover, such indices have to be site specific because they are integrated with local epidemiological data (Stieb et al. 2005). Won’t this confuse even more? Line 352: (2014: 1127) Reference is missing. Lines 342-353. Authors need to understand and review the state of science regarding sensors. There are a huge range regarding these sensors, and their accuracy and precision. All sensors may be small ‘Handheld devices’, but not all ‘handheld devices’ use the sensors that the authors imply in the manuscript. Lines 411-413: Additionally, how can one know the accuracy of these posts?
Author Response
I want to thank all the reviewers for their very constructive critique of my article. It was very fulfilling to hear positive feedback from two atmospheric chemists and an expert on AQI who recognize that social science can also play a productive and collaborative role in the measurement of particulate matter. It is my hope that this conceptual article can help stimulate new ways of measuring particulate matter that simultaneously increases awareness and action among those exposed to this pollution.
Reviewer 2 is correct that the original manuscript was not strictly a review of AQI. It was stated clearly in both the title, abstract and throughout the original manuscript that this article is about the relationship between both the measurement of particulate matter and the creation of air quality indices. For instance, the abstract stated that the article “reviews the relationship between discrete measurements and indices while also speculating on the way continuous measurement of air pollution could stimulate awareness and action.” That said, at many places my review of AQI in Section 3 of the original manuscript was a discussion of the Pros and Cons of current indices. For instance, at Line 218-220 the original manuscript states: “Air Quality Index (AQI) is necessary to transmit information about the health effects of air pollution to the public effectively and to support the development of proper policy to reduce air pollution.” The article then goes on to point out the complexity of calculating the index using examples from the U.S., China, Hong Kong and Europe that can actual serve to confuse the public. Following the comments of the other two reviewers for this article, the new manuscript will continue to include a discussion of the measurement of particulate matter. However, I have made the following corrections according to Reviewer 2’s concrete comments to make sure that the article and the review of AQI is done in a more proper manner.
Line 21: I have extended the acronym in the first instance.
Line 23: I have extended the acronym in the first instance.
Line 25: The online comments about APEC Blue are referenced in Zheng (2014) which is cited in the following sentence, thus the reference serves as documentation for both of these sentences.
Line 35-36: This sentence is the beginning of a new paragraph and the “we” does not refer to the government, but to society.
Line 42: This is a stylistic difference in some social science research. In single authored articles it is generally appropriate to use the singular pronoun.
Line 74: I have changed this sentence so that it is clearer that I am talking about particulates in the air.
Line 89: I have extended the acronym.
Line 100-103: I have removed this sentence at the request of another reviewer.
Line 111: I have added a space between fiber and filter.
Line 116: The reference to the relationship of density and aerodynamic diameter is located on page 59. For clarity, I have added that page number to the reference.
Line 117: I have removed “of course”.
Line 129-132: I have removed this sentence.
Line 147-150: I thank the reviewer for pointing out this sentence. There was a mistake here which I have corrected and I have also included the references for clarity.
Line 148: I have made the change to terminology here.
Line 150-153: I have changed this sentence to note that filter and optical methods are used in tandem.
Line 154: I have removed “of course”. I also rewrote this sentence for clarity.
Line 154-176: In this section I have made it clearer that GC/MS is used to determine the molecular composition of the particulates. I have also included a sentence at the end of this paragraph that mentions new technologies that are being developed to measure the chemical composition of particulates in real-time.
Line 177-178: This sentence does not imply that fungal spores are harmless, but I have removed “non-toxic” for clarity. I have changed toxic particles to toxic components.
Line 181: I have added incomplete
Line 190-192: I have removed “all”. The reference of Brook et al. 2010 discusses the way particulates can increase instances of heart disease.
Line 193: I have fixed this statement.
Line 204: I have removed et al. I have added an epidemiological reference that further supports the CDC citation already provided in this sentence.
Line 206: All three of the references provided in the original manuscript discuss and provide evidence of the reduction of funding for helping Latino communities cope with asthma. These references do not refer to reductions in African American communities, nor am I aware of any social science studies that have examined this issue.
Line 209-211: This sentence is logically sound with regard to the previous sentences. Scientific studies found that asthma was less prevalent among Latino communities and politicians reduced funding for those communities. As the three studies cited argue, these communities in fact need that support because they are “prone to high exposure of air pollutants” and thus information about air pollution and human health needs to be made more relevant to the needs of individual communities to prevent further unfair allocation of funds for helping communities cope with asthma.
Line 233, 245, 279: I fixed this typo.
Line 250: I have added the date, although I may be off by a day or two because I did not record the exact date I wrote this sentence, but it would have been written around August 23rd 2016 when I first developed this section.
Eq. 2: I have changed the hyphen.
Line 307: The reviewer is correct, this sentence needed to be reworded. However, the sentence does not imply that the health impacts of air pollution are inconsequential, but rather that these impacts were discussed more transparently in France, Germany and UK and therefor rarely become a controversial political issue as them might become in other parts of the EU. I have changed the wording to reflect this.
Line 313-314: The trend I am highlighting with this statement is that some indices make the connection between air quality and health more explicit. AQI does not do this, but AQHI does.
Line 317-318: As the next sentence states, integrating epidemiological data within an index is contentious because of generalizing assumptions that are typically made about the population, from which the epidemiological data is collected. The point of this summary is that integrating health impacts within an air quality index has pros and cons. Thus, here again we can see that the original manuscript already included a discussion of the pros and cons of different air quality indices that Reviewer 2 rightly said is important for Section 3.
Line 352: Holstius et al. 2014 was included in the references of the original manuscript.
Line 342-353: At the request of another reviewer I have added additional information about several specific sensors currently on the market.
Line 411-413: Schmitt and Li (Under Review) specifically talks about the fact that these posts are unverifiable and therefore they constitute a form of truthiness. As I note in the following sentence, these are the potential negative aspects that could emerge from the speculative methods I describe in this section.
Reviewer 3 Report
I am not a social scientist and am generally not familiar with this or similar journals. As an aerosol/atmospheric scientist and engineer, I found the general paper to be interesting and well-written but it is certainly not a research study. It is also not a meta-analysis of current literature although it does seem well-referenced and the author appears well-informed about both social science and technical aspects in this area. The paper provides an interesting idea/perspective on how to improve society's understanding an behavior/reactions to air quality issues but it does not offer a substantially specific recommendation for action. This is not necessarily a fault, but rather an observation. The paper does a good job explaining the parts of the science related to air quality rating systems and pollutants but not necessarily HOW/WHAT is measured by discrete (filter) or continuous (optical methods) air measurements. The title is very odd and not as connected to the content as it should be in an apparent attempt to be clever rather than descriptive. The comparisons to quantum theory and measurement are misleading and unnecessary and should be removed. They imply (or at least lead the reader to infer) that somehow we are not able to be accurate in air pollutant measurements or that somehow they are up to interpretation or debate. This is not the case. There are several places where the issues of calibration of equipment are mentioned in an attempt to also imply that these systems are not accurate or can vary wildly across methods. This is also not accurate.
What does Piercian mean? I have not heard this term and even looking it up is vague and confusing. The conclusion section offers many more questions than it answers and never proposes WHO or HOW such changes could be made. While it is true the engaging the public more and involving them in measurements and showing them how AQI values are calculated is a good goal, it does not even propose a first step (let alone an outlined plan) on how this might reasonable be achieved and who should be leading this cause. Why are only personal-portable monitors discussed rather than more permanently-placed local networks of air monitoring stations? Why are the specific technologies and brands currently on the market not mentioned? There are several small monitoring devices in tests around the world at this time and those systems and their advantages and limitations would add a great deal to this paper.
Author Response
I want to thank all the reviewers for their very constructive critique of my article. It was very fulfilling to hear positive feedback from atmospheric chemists and experts on AQI who recognize that social science can also play a productive and collaborative role in the measurement of particulate matter. It is my hope that this conceptual article can help stimulate new ways of measuring particulate matter that simultaneously increases awareness and action among those exposed to this pollution.
I want to first thank reviewer 3 for pointing out that this paper could benefit greatly by establishing a preliminary roadmap for how to integrate discrete and continuous measurements with public perceptions and actions. In section 4 I have outlined this roadmap and made suggestions for who could help place it into motion. I have also tried to cautiously reflect on the potential pitfalls that may emerge if these people were to place such a roadmap into action.
Additionally, in response to another reviewer, in section 2 I have clarified how and what is being measured by either filter or optical technologies.
I agree with the reviewer that the title was clever rather than descriptive, which is a common convention in the social sciences and humanities. It was intended to be a play on Gregory Bateson’s definition of information as “a difference that makes a difference”. I have changed the main title to Measuring Particulates and Inventing Indices, which more accurately describes the content of the paper.
My goal in drawing on quantum physics is not to suggest that measurements of particulate matter cannot be accurate, but rather to engage with Karen Barad’s influential concept of agential realism. However, I would argue that scientific debate about the way we make measurements is exactly what drives us to make ever more robust measurements. In section 2, I have made several additions and adjusted the way quantum physics is discussed in relation to the measurement of particulate matter. I also removed a sentence at Line 171-174 of the original manuscript to reduce the impression that calibration makes such measurements inaccurate.
A Peircian index refers to a specific form of linguistic structure that was developed by Charles Sanders Peirce. I have changed this to “symbolic index” for ease of understanding across disciplines.
According to the reviewer’s suggestion, I have added a detailed paragraph in section 4 regarding some of the benefits and downsides of several popular handheld devices. Most of the discussion of stationary monitoring devices remains within section 2, however section 4 also touches on community-based continuous monitoring systems which use stationary monitoring devices.
Round 2
Reviewer 2 Report
Page 1, Introductory paragraph
Can you mention or acknowledge occurrences of several other events similar to APEC events e.g. Beijing Olympics, Military parade events, etc.? Currently, it looks like APEC is the only time such intervention was made to improve air quality.
Page 2, Line 48: as well as how we represent those measurements to he public.
è to ‘the’ public.
Page 2, line 55: communication found in new media
è new media or news media?
Page 2, lines 72-72: but rather that measurements taken from multiple perspectives can help us improve our accuracy.
è Need more explanation on multiple perspectives and how it can improve accuracy. Multiple measurements can tell you about precision, but not necessarily about accuracy.
Page 3, line 121: concentration and chemical composition,
è I suggest to add size and number.
Page 4, lines 180-198: GC-MS is one of the instrumental techniques. There’re various other instruments that are routinely used for analyzing aerosol chemical composition.
Page 5, line 204: Aerodyne Aerosol Mass Spectrometery
è This instrument has been in market for over a decade and should not be acknowledged as such.
Page 9, lines 376-387: It’s Dylos, not ‘Dyson’.
Page 11, line 459: pDR-1500 is a portable instrument, and other appropriate instruments for stationary monitoring system should be mentioned.
Author Response
Many thanks again for pointing out some of the deficiencies of this paper. I have made the following specific corrections:
Introductory Paragraph: I have added a sentence and included a recent publication to support this point.
Line 48: I fixed the typo
Line 55: this should be "new media", it is referring to the digital media technologies discussed at the end of the article
Line 72: I have added a more clear description of what is meant here by accuracy and included a citation to support this statement.
Line 121: It is not clear what the reviewer is referring to here. The previous paragraph about the dimension of particulates is exactly about size. The following discussion about concentration is about the number of particulates.
Line 180-198: In the 2nd draft of this manuscript I added Aerodyne Aerosol Mass Spectrometery
as an additional method for analyzing aerosol chemical composition.
Line 204: I have removed the phrase "currently being developed".
Line 376-387 I have changed Dyson to Dylos
Line 459: I have removed the word "stationary" and added the TEOM-FDMS as a potential stationary system that could be utilized.
Reviewer 3 Report
This paper is been very much improved by the changes.
Author Response
Many thanks for your support